# Effect of Mandibular Advancement Device Treatment on the Site-Specific Degree of Upper Airway Collapse During Drug-Induced Sleep Endoscopy [note 1]

**DOI:** 10.3390/jcm14228142

**Published:** 2025-11-17

**Authors:** Eldar Tukanov, Marijke Dieltjens, Annelies E. Verbruggen, Anneclaire V. Vroegop, Johan A. Verbraecken, Paul H. Van de Heyning, Marc J. Braem, Sara Op de Beeck, Olivier M. Vanderveken

**Affiliations:** 1Translational Neurosciences, Faculty of Medicine and Health Sciences, University of Antwerp, 2610 Wilrijk, Belgium; 2Department of ENT, Head and Neck Surgery, Antwerp University Hospital, 2650 Edegem, Belgium; 3Multidisciplinary Sleep Disorders Centre, Antwerp University Hospital, 2650 Edegem, Belgium; 4Department of Pulmonary Medicine, Antwerp University Hospital, 2650 Edegem, Belgium; 5Research Group LEMP, Faculty of Medicine and Health Sciences, University of Antwerp, 2610 Wilrijk, Belgium

**Keywords:** mandibular advancement device, obstructive sleep apnea, sleep endoscopy, upper airway collapse, OSA, MAD, DISE

## Abstract

**Background:** Mandibular advancement device (MAD) therapy is a non-invasive treatment for obstructive sleep apnea (OSA). Although the effect of MAD on OSA outcomes is widely known, its effect on the upper airway collapse degree remains poorly understood. This study aimed to assess the impact of MAD therapy on site-specific airway collapse degree during drug-induced sleep endoscopy (DISE). **Methods:** One hundred participants were recruited and underwent a baseline polysomnography. Overall, 69 participants with OSA (AHI 5–50 events/h) underwent DISE at baseline and with MAD set to 75% of maximal mandibular protrusion. Collapse degree (none, partial, complete) was evaluated at the palate, oropharynx, tongue base, hypopharynx, and epiglottis without and with MAD. Ordinal logistic regression was used to analyze changes in degree of collapse. **Results:** MAD therapy reduced collapse degree at the palate (OR = 5.92 [3.28; 10.66], *p* < 0.001), oropharynx (OR = 2.70 [1.48; 4.92], *p* = 0.001), tongue base (OR = 1.83 [1.32; 2.53], *p* < 0.001), and hypopharynx (OR = 2.90 [1.53; 5.48, *p* = 0.001), with no effect at the epiglottis (OR = 0.65 [0.42; 1.02], *p* = 0.058). **Conclusions:** MAD therapy reduces upper airway collapse at most anatomical levels, except at the level of the epiglottis. These findings confirm its therapeutic efficacy while underscoring the importance of identifying patients with residual or worsening collapse who may benefit from combined or alternative treatments.

## 1. Introduction

Obstructive sleep apnea (OSA) is a chronic respiratory sleep disorder with a high prevalence, affecting up to 49% of men and 23% of women between 39 and 75 years old [1]. The clinical significance of OSA and its severity is highlighted by its important association with multiple comorbidities, making efficacious treatment mandatory [2,3,4,5,6]. In clinical practice, continuous positive airway pressure (CPAP) is the standard treatment for OSA, which opens the upper airway by creating a pneumatic splint [7]. Alternative treatment modalities, such as mandibular advancement device (MAD) therapy, have emerged as non-invasive options for patients with mild to moderate—and even severe—OSA or those intolerant to CPAP [8,9,10,11,12]. MAD therapy acts by advancing the mandible forward during sleep, thereby increasing the cross-sectional area at the levels of the upper airway and reducing the likelihood of airway collapse [13]. This treatment has proven to be effective in patients with mild to moderate OSA and comes with high adherence [12,14,15]. Furthermore, multiple studies showed a high efficacy in patients with severe OSA [16,17,18,19]. However, treatment response remains patient-dependent [20,21].

As OSA is a multifactorial disease with multiple possible mechanisms [22,23,24], further evaluation is often necessary to maximize treatment response [25]. Drug-induced sleep endoscopy (DISE) is currently used in routine clinical practice to assess the site(s), degree, and patterns of upper airway collapse in OSA [26]. During DISE, the upper airway is visualized during mimicked sleep, typically induced by propofol and midazolam, but other protocols with different sedatives may be used [27]. Research has already shown that there are some associations between MAD therapy outcome and the site, pattern, and degree of collapse during DISE [28,29,30]. However, less research has been performed concerning the effect of MAD therapy on the degree of collapse during drug-induced sleep.

The aim of this study is to assess the effectiveness of MAD in reducing the degree of collapse during DISE.

## 2. Materials and Methods

This study presents a secondary analysis of a prospective trial registered at clinicaltrials.gov (NCT01532050) [31]. The trial was approved by the local ethical committee at the University of Antwerp and Antwerp University Hospital. Written informed consent was obtained from all participants. As the current study is a secondary analysis, no additional patient inclusion or intervention was performed, and all analyses were performed on existing data collected during the original trial.

The trial protocol has been published previously by Verbruggen et al. (2016) (Figure 1A) [31]. One hundred participants were consecutively recruited by a multidisciplinary team, which included a dental sleep professional; an ear, nose, and throat (ENT) surgeon; and a board-certified sleep specialist responsible for patient care. During recruitment, participants underwent extensive screening per protocol as well as a comprehensive clinical evaluation conducted by an ENT surgeon and dental sleep specialist. Evaluation for temporomandibular joint issues encompassed anamnestic assessment, palpation, and functional appraisal of mandibular movements. Subsequently, a baseline assessment was conducted using full-night type 1 PSG to reconfirm eligibility criteria (Table 1) [32,33]. Afterwards, oral appliance therapy was initiated using a titratable, custom-made, duoblock MAD (Respident Butterfly MAD, Orthodontics Clinics NV, Antwerp, Belgium), adjusted to 75% of the individualized maximal mandibular protrusion, in which 100% is measured as the distance between maximal protrusion and maximal retrusion. This allowed standardization between patients while respecting the individual protrusion range of each patient. The maximum protrusive capacity of each participant was measured three times and averaged using a proprietary gauge bite fork (The BiteFix, Sheu-Dental, Iserlohn, Germany). Measurements were made according to the trajectory of the centric relation position to maximal protrusion. All PSG variables were scored by two sleep laboratory technicians dedicated to this study in accordance with the American Academy of Sleep Medicine criteria [34]. After initiating MAD therapy, all participants underwent a DISE between one and three months after MAD fitting. All DISE findings were recorded and re-evaluated through consensus scoring by a panel of four experienced ENT surgeons to minimize potential interobserver variability and ensuring robust scoring [35]. Scoring was performed by all four ENT surgeons simultaneously in the same room, without any individual scoring. The investigators and participants remained blinded during the data collection.

### 2.1. Drug-Induced Sleep Endoscopy

Participants underwent a DISE with and without MAD in situ. The DISE procedures took place in a semi-dark and quiet operating theater, with the participant positioned in a supine position. The investigation was performed by an experienced ENT surgeon and scored by a board of four experienced ENT surgeons to provide a consensus scoring of the results. To mimic natural sleep, a combination of sedatives was used. Sleep was initiated using an intravenous bolus administration of midazolam (1.5 mg) and maintained using target-controlled infusion of propofol (2.0–3.0 μg/mL). During the procedure, standard cardiovascular monitoring was carried out. The level of sedation was continuously assessed using bispectral index (BIS) monitoring (BIS VISTA monitor; Aspect Medical Systems Inc., Norwood, MA, USA) which involved a leaf of four sensor electrodes (BIS Quatro; Aspect Medical Systems Inc., Norwood, MA, USA) attached to the forehead. It recorded values between 0 (no brain activity) and 100 (fully awake) [36]. DISE assessment in the PROMAD study protocol was conducted at BIS values between 50 and 70. A flexible fiberoptic nasopharyngoscope (Olympus END-GP, diameter 3.7 mm, Olympus Europe GmbH, Hamburg, Germany) was inserted intranasally to inspect the upper airway. The degree (none, partial, or complete), direction (anteroposterior, concentric, or lateral), and level of collapse were evaluated using a modified VOTE classification, which also takes the level of the hypopharynx into consideration [31,37]. The following upper airway levels were examined: soft palate, oropharynx (region at the level of the tonsils), tongue base, epiglottis, and hypopharyngeal lateral walls (region below tongue base). In the current analysis, the different patterns for each collapse site were pooled with the analysis focusing on degree of collapse.

### 2.2. Statistical Analyses

All statistical analyses were performed using IBM SPSS Statistics 29 (IBM Corp, Armonk, NY, USA). As the primary analysis of this study, an ordinal logistic regression model with generalized estimating equations was utilized to evaluate the change of upper airway collapse degree (none, partial, or complete) from baseline to MAD, across different collapse sites (velum, oropharynx, tongue base, hypopharynx, epiglottis) during DISE. The model was based on the following formula for each specific collapse site:$$\log\left(\frac{P\left(collDegree\leq j\right)}{1-P\left(collDegree\leq j\right)}\right)\sim \alpha_{j}+\beta \left(treatment\right),\;\;j=1,2$$
*with treatment = 0 at baseline; treatment = 1 with MAD; and**with j = boundaries for the three categories (1 = none vs. partial collapse; 2 = partial vs. complete collapse).*

Analyses were conducted for the total population. The reported odds ratios (OR (95% confidence interval)) represent the likelihood of experiencing a reduced degree of collapse with MAD compared to the baseline. To assess the robustness of the observed effect of MAD treatment on the degree of upper airway collapse, additional sensitivity analyses were conducted adjusting the ordinal logistic regression model for potential confounders, including BMI and baseline AHI.

Additional analyses were considered exploratory: descriptive statistics were reported as mean ± standard deviation, median (quartile 1–quartile 2), or number and percentages. Normality was tested using the Shapiro–Wilk test. Normally distributed continuous variables were compared using one-way ANOVA, non-normally distributed variables using the Kruskal–Wallis test. To assess whether changes in site-specific upper airway collapse degree from baseline to MAD were associated with improvement in AHI, ODI, or mean SaO_2_, comparison of improvement in these parameters across three collapse change groups (improved collapse degree, no change, worsened collapse degree) for each anatomical site was performed using the non-parametric Kruskal–Wallis test. To compare categorical variables between the two OSA severity subgroups (mild and moderate-to-severe), exploratory analyses using the Pearson’s chi-square test or Fisher–Freeman–Halton Exact test were performed. An ordinal logistic regression model with generalized estimating equations was utilized separately for subgroups categorized by OSA severity (mild and moderate-to-severe). Descriptive statistics, normality tests, and ordinal logistic regression models with generalized estimating equations were repeated specifically for three OSA severity subgroups (mild, moderate, severe) and can be found in the Appendix A. Additional sensitivity analyses were performed using the original VOTE classification instead of the modified version [37]. For these analyses, the scoring at the oropharynx and hypopharynx were combined, and the highest collapse degree was kept. The *p*-values were reported as two-sided, and statistical significance was established at *p* < 0.05. Adjustments for multiple comparisons were not made.

## 3. Results

A total of 100 participants with OSA (83% male; mean ± SD age: 47.6 ± 10.0 years; mean ± SD BMI: 26.9 ± 3.3 kg/m^2^; median (Q1–Q3) AHI: 14.6 (9.4–23.8) events/h sleep) were prospectively included for this study and received a new baseline PSG. One participant did not complete this PSG due to time constraints, ten participants showed AHI < 5 events/h, and three participants showed AHI > 50 events/h. This resulted in a total of 86 out of 100 participants being eligible for further participation. Overall, 71 participants underwent DISE, both with and without MAD. DISE results were marked as unclear in two participants, resulting in a final dataset of sixty-nine participants with both a baseline PSG and a DISE with and without MAD (Figure 1B). Baseline clinical and demographical characteristics of all 69 participants can be found in Table 2. The VAS scores for snoring and minimal oxygen saturation were not available in one and two participants with AHI < 15, respectively. Additional clinical and demographical characteristics of the population at baseline, in which the moderate-to-severe group (15/h ≤ AHI < 50/h) has been subdivided in a moderate OSA group (15/h ≤ AHI < 30/h) and severe OSA group (30/h ≤ AHI < 50/h), can be found in Appendix A.

### 3.1. Population Characteristics at Baseline

A visual representation on the distribution of collapse degree (none, partial, or complete) at baseline can be found in Figure 2. This figure shows the collapse distributions across all five anatomical levels (palate, oropharynx, tongue base, hypopharynx, and epiglottis) for the total subgroup (Figure 2A) and for specific OSA severity subgroups: mild (Figure 2B), moderate-to-severe (Figure 2C). Across all subgroups, palatal collapse was the most common site of collapse, with 28 participants of the total population having partial collapse and 37 participants having complete collapse at this level. At the oropharynx, seven participants had complete collapse and sixteen had partial collapse. At the tongue base six participants had complete collapse and thirty had partial collapse. At the level of the hypopharynx, only partial (total population: 17 participants) or no collapse (total population: 52 participants) was observed. At the level of the epiglottis, collapse (partial or complete) was only observed in thirteen participants (six complete, seven partial). When looking at the distributions based on OSA severity (Figure 2B,C), statistical analysis using the Fisher–Freeman–Halton Exact test showed that there were no significant differences in the distribution of collapse degree across OSA severity subgroups for any anatomical level: *p* = 0.139 at the palate; *p* = 0.400 at the oropharynx; *p* = 0.449 at the tongue base; *p* = 0.265 at the hypopharynx; and *p* = 0.905 at the epiglottis.

An additional visual distribution of collapse degree at baseline in which the moderate-to-severe group (15/h ≤ AHI < 50/h) has been subdivided into a moderate OSA group (15/h ≤ AHI < 30/h) and a severe OSA group (30/h ≤ AHI < 50/h) can be found in Appendix A.

### 3.2. Population Characteristics During MAD Therapy

Figure 3 illustrates the distribution of collapse degree (none, partial, or complete) with MAD therapy across all five anatomical levels (palate, oropharynx, tongue base, hypopharynx, and epiglottis) for the total cohort (Figure 3A) and for OSA severity subgroups: mild (Figure 3B), moderate-to-severe (Figure 3C). In the total population, there was a shift towards a lower degree of collapse with MAD. Interestingly, the number of participants having no collapse at the level of the epiglottis was lower with MAD for all OSA severity subgroups. In the total population, there was no collapse at this level at baseline in 56 participants, and this number decreased to 50 participants with MAD. In the mild OSA population, there was a decrease from 27 participants at baseline to 26 participants with MAD. For moderate-to-severe OSA, there was a decrease from 29 participants at baseline to 24 participants with MAD. When looking further at the distributions with MAD based on OSA severity (Figure 3B,C), statistical analysis using the Fisher–Freeman–Halton Exact test showed that there were no significant differences in the distribution of collapse degree across OSA severity subgroups for any anatomical level: *p* = 1.000 at the palate; *p* = 0.184 at the oropharynx; *p* = 1.000 at the tongue base; *p* = 0.106 at the hypopharynx; and *p* = 0.781 at the epiglottis.

An additional visual distribution of collapse degree with MAD therapy in which the moderate-to-severe group (15/h ≤ AHI < 50/h) has been subdivided in a moderate OSA group (15/h ≤ AHI < 30/h) and a severe OSA group (30/h ≤ AHI < 50/h) can be found in Appendix A.

### 3.3. Regression Analysis

Ordinal logistic regression analysis (Table 3) assessed the change of upper airway collapse degree from baseline to MAD for each specific site of collapse during DISE in the total population (Figure 4). There were significantly high odds for having a lower degree of collapse at the palate (OR = 5.92 [3.28; 10.66], *p* < 0.001), oropharynx (OR = 2.70 [1.48; 4.92], *p* = 0.001), tongue base (OR = 1.83 [1.32; 2.53], *p* < 0.001), and hypopharynx (OR = 2.90 [1.53; 5.48], *p* = 0.001). The effect was most pronounced at the palatal level, with an odds ratio indicating participants were nearly six times more likely to have a lower collapse degree at this level with MAD therapy. No significant effect was observed at the epiglottis in the total cohort (OR = 0.65 [0.42; 1.02], *p* = 0.058).

To assess the robustness of the observed effect of MAD treatment on the degree of upper airway collapse, sensitivity analyses were conducted adjusting the ordinal logistic regression model for potential confounders, including BMI and baseline AHI (Table 3). When both covariates were added to the model, the effect of MAD therapy on the degree of collapse showed no significantly different results.

### 3.4. Exploratory Analyses

To assess whether changes in site-specific upper airway collapse degree following MAD therapy were associated with improvement in AHI, a non-parametric comparison of AHI improvement across three collapse change groups (improved collapse degree, no change, worsened collapse degree) for each anatomical site was performed. Across all collapse sites, no statistically significant differences in AHI improvement were observed between the collapse change groups (all *p* > 0.05): palate (*p* = 0.140); oropharynx (*p* = 0.656); tongue base (*p* = 0.388); hypopharynx (*p* = 0.833); and epiglottis (*p* = 0.400). This was extended to ODI improvement and improvement in mean SaO_2_, which all showed no statistically significant differences in improvement of these parameters between the three groups.

As another exploratory analysis, the original ordinal logistic regression analysis was repeated in the mild and moderate-to-severe OSA subgroups (Table 4). In the mild OSA subgroup (AHI < 15), significantly high odds for a lower collapse degree with MAD were observed at the palate (OR = 7.70, *p* < 0.001), oropharynx (OR = 3.63, *p* = 0.035), tongue base (OR = 1.56, *p* = 0.034), and hypopharynx (OR = 7.07, *p* = 0.038). In the moderate-to-severe OSA subgroup (15 ≤ AHI < 50), this was also true for the palate (OR = 4.82, *p* < 0.001), oropharynx (OR = 2.54, *p* = 0.014), tongue base (OR = 2.24, *p* = 0.003), and hypopharynx (OR = 2.22, *p* = 0.019). There were low odds for a lower collapse degree with MAD at the level of the epiglottis in moderate-to-severe OSA patients (OR = 0.49, *p* = 0.043). Importantly, only six participants had any form of epiglottis collapse to begin with at baseline, which increased to nine participants.

To assess the robustness of the observed effects, sensitivity analyses were conducted adjusting the ordinal logistic regression model for potential confounders, including BMI and baseline AHI (Table 4). When both covariates were added to the model, the effect of MAD therapy on the degree of collapse in the subgroups (mild and moderate-to-severe) showed no significantly different results.

The ordinal logistic regression and sensitivity analyses were repeated for the three OSA subgroups, in which the moderate-to-severe group (15/h ≤ AHI < 50/h) has been subdivided into a moderate OSA group (15/h ≤ AHI < 30/h) and a severe OSA group (30/h ≤ AHI < 50/h). The results can be found in Appendix A. Additionally, all analyses were repeated using the original VOTE classification, instead of the modified version that included the hypopharyngeal site, to assess the robustness of our findings to the definition of collapse classification (Appendix A). The results remained consistent, with comparable direction and magnitude of regression coefficients.

## 4. Discussion

The findings of this study provide valuable insights into the effects of MAD therapy on upper airway dynamics in participants with OSA. Overall, MAD therapy significantly improved collapse degree across most anatomical levels, except at the level of the epiglottis.

The most pronounced effect was observed at the level of the palate, with a significant reduction in collapse across the total population and all OSA severity groups. A previous analysis of this study population found that complete concentric collapse of the palate (CCC) during DISE at baseline was associated with poorer MAD treatment outcome [38]. Our results demonstrate a significant effect of MAD at the palate. Although both findings are not directly correlated, it might be possible that MAD therapy still has an effect on CCC. Future research taking into account collapse direction and focusing on patients with CCC is needed to confirm these findings.

At the oropharynx and tongue base, a significant reduction in collapse was observed in the overall population as well as in the OSA subgroups. Previous research showed mixed results regarding oropharyngeal collapse as a predictor of successful treatment outcomes, while a prior analysis of this study population revealed an association between tongue base collapse and favorable MAD therapy results [38,39]. Our findings might help in understanding the underlying mechanisms of these results, but further research in a larger population is necessary to validate this statement.

Hypopharyngeal collapse also decreased in the overall population and both OSA subgroups. Importantly, only a limited number of participants (n = 17) had collapse at this level to begin with, and none of them had complete hypopharyngeal collapse. As the hypopharynx is not scored in the commonly used VOTE classification, research on this collapse level is limited [37].

Interestingly, the epiglottis showed the least improvement with MAD therapy across the total population and mild subgroup, with even some worsening in the moderate-to-severe subgroup, while prior studies report a favorable response of epiglottic collapse to MAD therapy [40,41]. As such, this might suggest that MAD treatment is not hampered by or does not alter epiglottic collapse and might even have a negative impact at this level. A possible explanation is that in moderate-to-severe OSA, the pathophysiological mechanisms differ from those in milder cases. Another consideration is that protrusion may inadvertently increase retroflexion of the epiglottis. This could explain the observed worsening in the moderate-to-severe group, where multi-level or more complex collapse patterns are more prevalent. Important to note is that only 13 out of 69 participants showed collapse at the level of the epiglottis at baseline, which is a small sample for drawing major conclusions. Especially in the moderate-to-severe OSA group (n = 35), only six participants showed any form of epiglottis collapse at baseline. As such, these findings should be validated using studies with larger samples.

A recent paper by Fernández-Sanjuán et al. (2024) described that mandibular advancement using an MAD-simulation device during DISE did not show any improvement of primary epiglottis collapse in mild and severe OSA patients [42]. Additionally, a small group of patients even showed deterioration at the epiglottis with mandibular advancement. On the other hand, secondary epiglottal collapses improved when tongue base collapse also improved with the use of MAD, and this in all severity groups. Although no distinction was made between primary and secondary epiglottic collapse in the current study due to the small sample size, the findings at the epiglottis are in line with the findings by Fernández-Sanjuán et al. (2024) [42]. Similar to our study, the paper also examined changes in upper airway collapsibility during DISE using a titratable mandibular advancement simulator, reporting reductions in collapsibility at the palate, oropharynx, and tongue base, and significant improvement across severity groups. While a simulation bite is titratable and mimics the position of an MAD, it has a lower retention during DISE, which may influence the observed collapse patterns. In contrast, the current study evaluated the therapeutic effect of a definitive MAD after 1 to 3 months of actual use. This distinction is crucial, as our participants had already undergone a period of adaptation to the device, and those unable to tolerate MAD therapy were excluded prior to DISE. Therefore, our findings reflect the impact of MAD as a treatment rather than a simulated maneuver. As such, the current study is the first study—to our knowledge—to specifically evaluate the effect of MAD therapy on the degree of upper airway collapse during DISE.

When dividing participants into groups in which collapse improved, remained unchanged, or worsened at any of the evaluated sites of collapse, no significant differences in AHI improvement were found between these groups. Expanding these analyses to ODI improvement and improvement to mean SaO_2_ also showed no significant differences in improvement between these groups. These findings suggest that site-specific changes in collapse degree, as assessed during DISE, may not directly predict MAD treatment response in terms of AHI, ODI, and mean SaO_2_ reduction. This could reflect the multifactorial nature of OSA pathophysiology, where improvements in clinical parameters may occur independently of visible changes in collapse patterns or may be influenced by other OSA endotypes such as collapsibility or muscle responsiveness [35,43]. Altogether, it shows that while DISE provides valuable anatomical and functional insights, its predictive value for treatment outcomes remains limited when used in isolation. Importantly, there is little power in this exploratory analysis due to the small samples in several groups, especially in the groups with worsening of collapse degree. Two studies have already assessed the associations between the DISE findings and treatment response in the population of this study [38,44]. Further studies with larger sample sizes are needed to clarify the relationship between anatomical collapse patterns and MAD treatment outcomes.

### Strengths and Limitations

Overall, the results of this study provide fundamental insight on the underlying mechanisms of MAD therapy in OSA patients. A key strength of this study is the prospective and consecutive inclusion of participants, significantly minimalizing any potential bias in participant recruitment. Additionally, DISE with and without MAD were performed during the same procedure for each participant, increasing the reliability of the DISE findings by reducing variability introduced by time or external conditions. Furthermore, the distribution of collapse degree at baseline did not significantly differ between OSA severity subgroups at any collapse level. This suggests that the subgroups were relatively homogeneous in terms of collapse degree, allowing for a fair comparison of the effect of MAD therapy across the different OSA severities.

As standardized guidelines for MAD titration are currently unavailable [45], a fixed protrusion setting of 75% of maximal protrusion was used in this study to ensure uniformity across participants. In contrast, clinical practice at our hospital typically involves individualized titration, optimized for each participant and generally ranging between 75% and 100% of maximal protrusion, which is likely to enhance treatment outcomes. However, to ensure consistency in outcome predictions for MAD therapy in this study, a uniform, fixed degree of protrusion was implemented [31].

Endoscopic assessments, such as DISE, are inherently subjective and prone to intra- and interobserver variability [46,47,48]. To mitigate this potential limitation, a standardized classification system was employed to score DISE findings [31]. All DISE findings were recorded and re-evaluated through consensus scoring by a panel of four experienced ENT surgeons to minimize potential interobserver variability and ensure robust scoring. The distribution of upper airway collapse degree across all levels is in line with the previous literature [48,49,50], which showed that palatal collapse is the most common site of collapse, followed by tongue base collapse. These findings suggest that the airway collapse characteristics of the study cohort closely reflect those of the general OSA population.

Despite its strengths, this study has several limitations. The relatively small sample size of sixty-nine participants, including only seven participants with severe OSA (AHI 30/h to 50/h), may limit the generalizability of the findings, especially when comparing the results between severity subgroups. Furthermore, only participants with an AHI between 5 and 50 events/h were included in the original trial, which influences results in the total population and severe OSA subgroup. The reason was that at the time of the study, there was a prevailing belief that MAD therapy was not effective in patients with severe OSA. Therefore, patients with an AHI > 50 were excluded from this study. This selection criterion is also reflected in the relatively low median ODI of 4.4/h and relatively high median mean saturation of 95.1% for the total population. As a result, the moderate and severe subgroups had to be combined into the ‘moderate-to-severe’ subgroup, with a total population of 35 participants. Combining these subgroups might affect the results and thus should be taken into account as a limitation during interpretation of statistical analyses. Furthermore, the small sample size limits the ability to draw definitive conclusions regarding collapse at the epiglottis, as only 13 of 69 participants showed epiglottic collapse at baseline. This restricted statistical power to analyze primary versus secondary epiglottic collapse. Future studies with larger and more diverse populations are needed to validate the present findings and further explore the long-term effects of MAD therapy on upper airway collapse and clinical outcomes.

The choice for using a fixed degree of mandibular advancement, namely, 75% of maximal protrusion, also has consequences. Although it does result in consistency throughout this study, it is possible that a greater degree of advancement may have altered the findings, as is usually conducted in clinical practice. In the end, both choices have their strengths and limitation.

Additionally, the current analysis did not take the collapse patterns—which are a crucial part of DISE scoring—into account, as this was beyond the scope of this paper. Future research should investigate the effect of MAD therapy on DISE collapse patterns to provide a more comprehensive understanding of its mechanisms. Another important limitation is the subjective nature of DISE scoring, which may introduce inter- and intra-rater variability. Although this issue was addressed during this study using consensus scoring by a panel of four experienced ENT doctors, the subjective nature of DISE scoring could result in different grading dependent on the scorer and even the time of scoring. Quantitative measurements of upper airway dimensions could offer a more objective assessment. Unfortunately, the DISE procedures were not carried out with a formal quantitative analysis in mind. Therefore, no reliable anatomical reference points to correct for endoscope position are present in the current dataset. As a result, reliably assessing changes in airway dimensions during DISE is challenging due to the difficulty in correcting for the position of the fiberoptic laryngoscope, especially in the palatal region where reliable anatomical reference points are necessary. Further research should specifically aim to investigate quantitative changes of upper airway dimensions during MAD treatment.

Data on neck circumference and retrognathia were not collected for the original trial. Consequently, these confounders could not be corrected for during regression analyses.

Lastly, DISE with MAD was performed three months after the initiation of MAD therapy. This treatment period could have influenced the observed findings, potentially confounding the interpretation of the immediate effects of MAD on upper airway dynamics.

## 5. Conclusions

MAD therapy effectively reduces the degree of airway collapse across most anatomical levels in participants with OSA, except at the level of the epiglottis. These findings confirm the therapeutic value of MAD therapy and its capacity to mitigate upper airway collapse degree at multiple sites, while also highlighting interindividual variability. Identifying patients with residual or worsening collapse remains essential to optimize treatment strategies, including consideration of combined or alternative approaches.

## Figures and Tables

**Figure 1 jcm-14-08142-f001:**
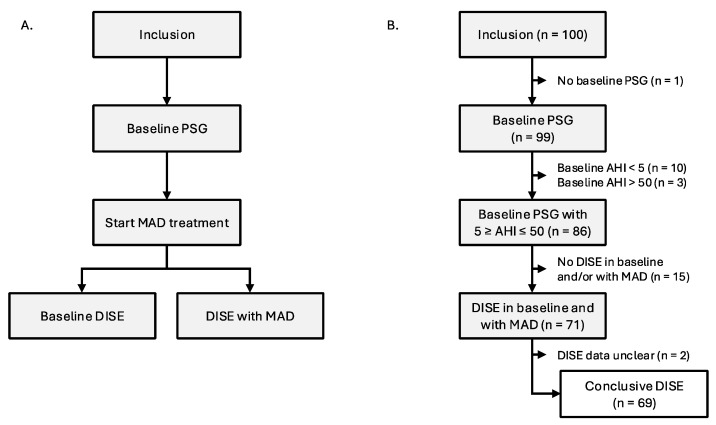
(**A**): Study flow. (**B**): Participant flow. Abbreviations: AHI (apnea–hypopnea index); DISE (drug-induced sleep endoscopy); MAD (mandibular advancement device); PSG (polysomnography).

**Figure 2 jcm-14-08142-f002:**
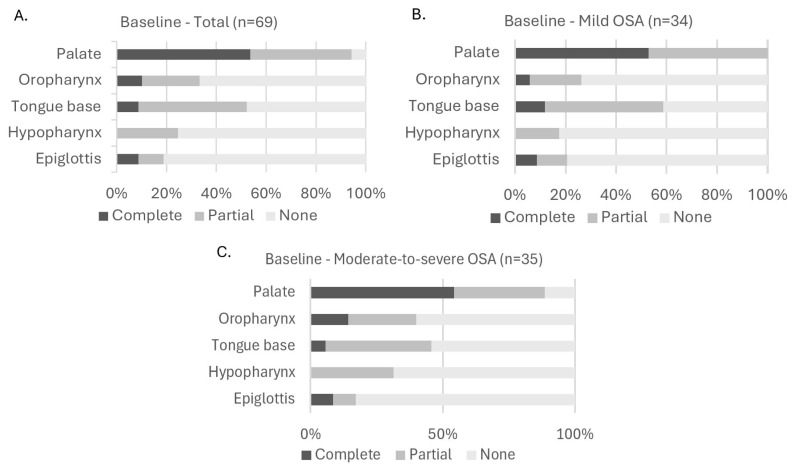
Distribution of collapse degree at each upper airway level at baseline. (**A**): Collapse in all participants; (**B**): collapse in participants with mild OSA; (**C**): collapse in participants with moderate-to-severe OSA. Fisher–Freeman–Halton Exact test showed no significant differences in collapse degree distribution across OSA severity subgroups (panel (**B**,**C**)). Abbreviations: OSA (obstructive sleep apnea).

**Figure 3 jcm-14-08142-f003:**
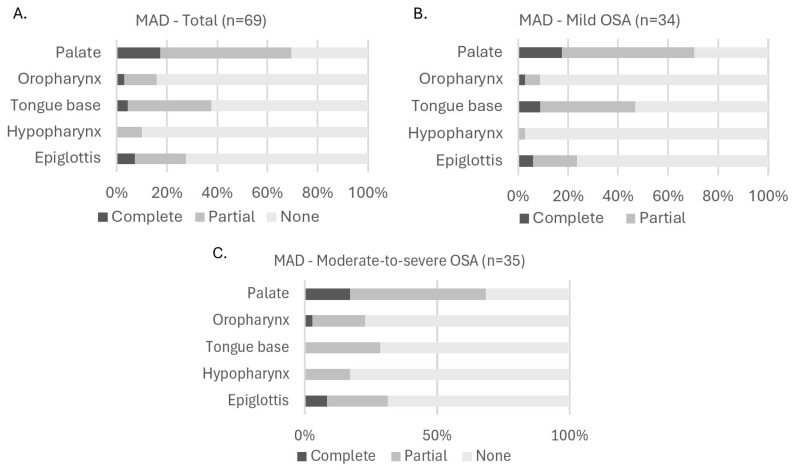
Distribution of collapse degree at each upper airway level with MAD. (**A**): Collapse in all participants; (**B**): collapse in participants with mild OSA; (**C**): collapse in participants with moderate-to-severe OSA. Fisher–Freeman–Halton Exact test showed no significant differences in collapse degree distribution across OSA severity subgroups (panel (**B**,**C**)). Abbreviations: MAD (mandibular advancement device); OSA (obstructive sleep apnea).

**Figure 4 jcm-14-08142-f004:**
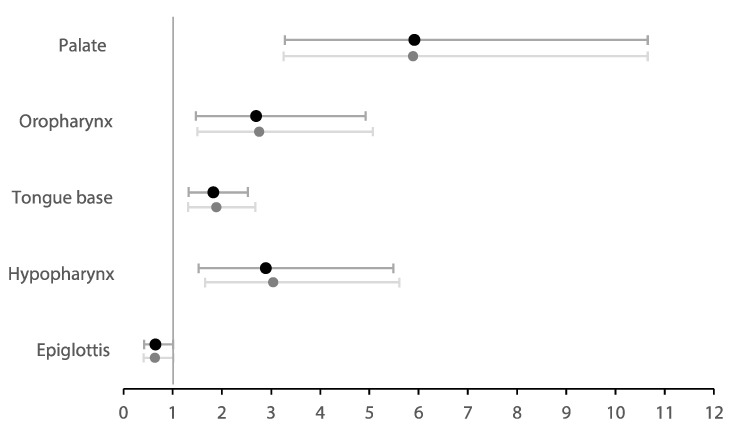
Forest plot showing the odds ratios of the change of site-specific upper airway collapse degree—from baseline to MAD therapy—in the total population. Dark plots show the original model; light plots show the model corrected for BMI and AHI.

**Table 1 jcm-14-08142-t001:** Eligibility criteria.

**Inclusion Criteria**
Age ≥ 18 years BMI ≤ 35 kg/m^2^ OSA as defined by the American Academy of Sleep Medicine task force [34] Diagnostic criteria: (A + B + D or C + D) ○ A. Anamnesis (at least one of the following criteria) ▪ Unwanted sleepiness and/or fatigue during the daytime, unrefreshing sleep, or insomnia ▪ Nocturnal arousals with breathing stops, gasping ▪ Snoring or breathing stops while sleeping, determined by the bed partner ○ B. PSG: AHI ≥ 5 events/h of sleep and AHI < 50 events/h of sleep ○ C. PSG: AHI ≥ 15 events/h of sleep and AHI < 50 events/h of sleep ○ D. The condition cannot be explained by another sleep disorder, internal or neurological disorder, medication, or drug use
**Exclusion criteria**
Absolute dental contra-indications: ○ Functional restrictions of the temporomandibular joint ○ Insufficient dentition with pathological aspects ○ Insufficient retention for MAD use Other sleep disorders (e.g., parasomnias) Previous invasive upper airway surgery for sleep-disordered breathing Genetic disorders with craniofacial and/or upper airway anomalies Use of benzodiazepine(s) and/or antidepressant(s) Prior history of psychiatric disease (including alcohol abuse) Known history of fibromyalgia or chronic fatigue syndrome Unwilling to participate and/or to give informed consent

Abbreviations: AHI (apnea–hypopnea index); BMI (body mass index); MAD (mandibular advancement device); OSA (obstructive sleep apnea); PSG (polysomnography).

**Table 2 jcm-14-08142-t002:** Clinical and demographical characteristics of population at baseline.

	Total (n = 69)	Mild OSA: AHI < 15 (n = 34)	Moderate-to-Severe OSA: 15 ≥ AHI < 50 (n = 35)	*p*-Value
Sex (male/female; n)	59/10	28/6	31/4	0.513 ^a^
Age (years)	48.4 ± 9.5	48.7 ± 11.1	48.1 ± 7.9	0.780 ^b^
BMI (kg/m^2^)	27.5 ± 3.1	26.7 ± 3.3	28.3 ± 2.6	0.025 ^b^
VAS (0–10) snoring	6.0 (5.0–9.0)	6.0 (5.0–9.0)	7.0 (6.0–9.0)	0.432 ^c^
ESS (0–24)	7.0 (5.0–13.5)	7.5 (4.0–11.0)	7.0 (5.0–15.0)	0.318 ^c^
AHI (events/h)	16.5 (11.1–23.5)	11.1 (7.3–12.8)	23.3 (19.2–29.0)	<0.001 ^c^
Supine AHI (events/h)	29.4 (18.3–52.5)	20.7 (12.3–36.0)	40.4 (28.8–54.6)	0.002 ^c^
Non-supine AHI (events/h)	8.9 (4.2–16.5)	5.9 (3.0–8.7)	15.7 (9.9–20.9)	<0.001 ^c^
ODI (events/h)	4.4 (2.4–10.9)	2.6 (1.2–4.5)	10.1 (3.8–15.5)	<0.001 ^c^
Mean SaO_2_ (%)	95.1 (94.1–96.1)	95.5 (94.4–96.4)	94.6 (93.6–95.7)	0.026 ^c^
Minimal SaO_2_ (%)	87.0 (84.0–90.0)	89.0 (84.6–91.0)	86.0 (83.0–88.3)	0.047 ^c^

Abbreviations: AHI (apnea–hypopnea index); BMI (body mass index); ESS (Epworth sleepiness scale); ODI (oxygen desaturation index); OSA (obstructive sleep apnea); SaO_2_ (oxygen saturation); VAS (visual analog scale for snoring). Note: Data are presented as median (quartile 1–quartile 3) for non-normally distributed data or mean ± SD for normally distributed data. AHI was scored according to the American Academy of Sleep Medicine 1999 criteria (3% oxygen desaturation or an arousal) [34]. ODI was calculated as dips of ≥3% over the total time in bed. ^a^ Fisher–Freeman–Halton Exact test; ^b^ ANOVA (analysis of variance); ^c^ Kruskal–Wallis test; *p* values compare the three subgroups.

**Table 3 jcm-14-08142-t003:** Logistic regression analysis on the change of site-specific upper airway collapse degree from baseline to MAD therapy. Without and with correction for BMI and AHI.

Total (n = 69)				
Site of Collapse	Ordinal Logistic Regression		+ BMI + AHI	
	*p*	OR (95% CI)	*p*	OR (95% CI)
Palate	<0.001	5.91 (3.28; 10.66)	<0.001	5.96 (3.29; 10.80)
Oropharynx	0.001	2.70 (1.48; 4.92)	<0.001	2.81 (1.52; 5.18)
Tongue base	<0.001	1.83 (1.32; 2.53)	<0.001	1.90 (1.33; 2.70)
Hypopharynx	0.001	2.90 (1.53; 5.48)	<0.001	3.17 (1.71; 5.88)
Epiglottis	0.058	0.65 (0.42; 1.02)	0.069	0.66 (0.42; 1.03)

Abbreviations: AHI (apnea–hypopnea index); BMI (body mass index); OR (odds ratio). Note: *p*-values and OR were calculated with a confidence interval of 95%.

**Table 4 jcm-14-08142-t004:** Exploratory logistic regression analysis on the change of site-specific upper airway collapse degree from baseline to MAD therapy, for each OSA severity group. Without and with correction for BMI and AHI.

**Mild OSA: AHI < 15 (n = 34)**				
**Site of Collapse**	**Ordinal Logistic Regression**		**+ BMI + AHI**	
	** *p* **	**OR (95% CI)**	** *p* **	**OR (95% CI)**
Palate	<0.001	7.70 (3.18; 18.64)	<0.001	7.78 (3.18; 19.06)
Oropharynx	0.035	3.63 (1.09; 12.05)	0.035	3.64 (1.09; 12.11)
Tongue base	0.034	1.56 (1.03; 2.37)	0.035	1.69 (1.04; 2.75)
Hypopharynx	0.038	7.07 (1.11; 44.89)	0.044	7.25 (1.06; 49.65)
Epiglottis	0.664	0.89 (0.52; 1.52)	0.755	0.92 (0.52; 1.60)
**Moderate-to-Severe OSA: 15 ≥ AHI < 50 (n = 35)**				
**Site of Collapse**	**Ordinal Logistic Regression**		**+ BMI + AHI**	
	** *p* **	**OR (95% CI)**	** *p* **	**OR (95% CI)**
Palate	<0.001	4.82 (22; 10.47)	<0.001	4.88 (2.25; 10.61)
Oropharynx	0.014	2.43 (1.20; 4.94)	0.007	2.64 (1.30; 5.36)
Tongue base	0.003	2.24 (1.32; 3.79)	0.003	2.24 (1.32; 3.80)
Hypopharynx	0.019	2.22 (1.14; 4.30)	0.006	2.90 (1.35; 6.19)
Epiglottis	0.043	0.49 (0.23; 0.98)	0.042	0.49 (0.24; 0.98)

Abbreviations: AHI (apnea–hypopnea index); BMI (body mass index); OR (odds ratio). Note: *p*-values and OR were calculated with a confidence interval of 95%.

## Data Availability

The data presented in this study are available on request from the corresponding author due to privacy restrictions.

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
