# Peer review of "Effect of Mandibular Advancement Device Treatment on the Site-Specific Degree of Upper Airway Collapse During Drug-Induced Sleep Endoscopyâ€"

_jcm, 2025, doi:10.3390/jcm14228142_

Round 1
Reviewer 1 Report
Comments and Suggestions for Authors
This manuscript is a prospective series aimed to evaluate the effect of mandibular advancement device (MAD) treatment on the site-specific degree of upper airway collapse during drug-induced sleep endoscopy.
The authors included 69 participants with OSA (AHI 5–50 events/h) that underwent DISE at baseline and with MAD set to 75% of maximal mandibular protrusion. They found that MAD therapy reduced collapse degree at the palate (OR = 5.92 [3.28; 10.66], p < 0.001), oropharynx (OR = 2.70 [1.48; 4.92], p = 0.001), tongue base (OR = 1.83 1.32; 2.53], p < 0.001), and hypopharynx (OR = 2.90 [1.53; 5.48, p = 0.001), with no effect at the epiglottis (OR = .65 [.42; 1.02], p = 0.058).
Thus, they concluded that MAD therapy reduces upper airway collapse at most anatomical levels, except at the level of the epiglottis.
It is an interesting and well-written work.
The manuscript addresses an important clinical question regarding the impact of MADs on site-specific upper airway collapse assessed with DISE. The methodology and statistical analysis are solid, and the study adds value by confirming findings in an independent cohort.
However, there are concerns that warrant a Major Revision:
Originality and prior literature:
The discussion presents the work as the first study to analyze MAD effects on different levels of collapse and in all severity subgroups. However, Fernández-Sanjuán et al. (2024) had already published a work on this topic and demonstrated that mandibular advancement reduces collapse at the palate, oropharynx, and tongue base, but does not resolve primary epiglottic collapse. Importantly, they were also the first to report that mandibular advancement can actually worsen epiglottic collapse in some patients.
Methodology:
- Lines 45–49: The epidemiological description of prevalence could be simplified.
- Line 63: It should be acknowledged that MAD efficacy has also been demonstrated in patients with severe OSA in multiple studies.
- Line 70: DISE induction can be performed with other drugs or protocols; this variability should be acknowledged.
- Line 428: Mixing moderate and severe patients in some analyses limits interpretation and should be explicitly stated as a limitation.
Results and interpretation:
- The finding of no effect at the epiglottis is important but incomplete: the authors should distinguish between primary collapse (not resolved by MAD) and secondary collapse (may improve).
- No significant correlation was found between changes in collapse and polysomnographic parameters (AHI, ODI, SaOâ‚‚). This point deserves a deeper discussion.
Conclusion:
The authors currently conclude that MAD reduces collapse at all levels except the epiglottis and may be recommended as a first-line treatment in moderate and severe OSA patients. However, this conclusion does not fully reflect their results, which show that residual collapses persist despite MAD use, that epiglottic collapse may even worsen in some cases, and that the elimination of collapses during DISE does not necessarily translate into improvements in polysomnographic parameters. A more accurate conclusion would be:
MAD therapy effectively reduces the degree of airway collapse across most anatomical levels in participants with OSA, except at the level of the epiglottis. While these results confirm the therapeutic value of MAD, they also emphasize the need to identify patients with residual or worsening collapse who may require combined or alternative treatments. Therefore, MAD should be considered an effective therapeutic option, particularly in carefully selected patients across the OSA severity spectrum, rather than a universal first-line treatment.”
Author Response
Originality and prior literature:
Comment 1: The discussion presents the work as the first study to analyze MAD effects on different levels of collapse and in all severity subgroups. However, Fernández-Sanjuán et al. (2024) had already published a work on this topic and demonstrated that mandibular advancement reduces collapse at the palate, oropharynx, and tongue base, but does not resolve primary epiglottic collapse. Importantly, they were also the first to report that mandibular advancement can actually worsen epiglottic collapse in some patients.
Response 1: Thank you for this valuable comment. We acknowledge the important contribution of Fernández-Sanjuán et al. (2024), which is cited in the Discussion section. It is important to note, however, that their study assessed the effect of a simulation bite rather than a definitive mandibular advancement device (MAD). While a simulation bite mimics the position of an MAD it has significantly lower retention and requires external force to maintain its position during DISE, which may influence the observed collapse patterns. In contrast, our study evaluated the therapeutic effect of a definitive MAD after 1 to 3 months of actual use. This distinction is crucial, as our participants had already undergone a period of adaptation to the device, and those unable to tolerate MAD therapy were excluded prior to DISE. Therefore, our findings reflect the impact of MAD as a treatment rather than a simulated maneuver. These differences are now clarified in the revised Discussion (lines 364-384).
Methodology:
Comment 2: Lines 45–49: The epidemiological description of prevalence could be simplified.
Response 2: Thank you for this comment. The description of OSA has now been simplified.
Comment 3: Line 63: It should be acknowledged that MAD efficacy has also been demonstrated in patients with severe OSA in multiple studies.
Response 3: Thank you for your comment. This has been added to the paragraph.
Comment 4: Line 70: DISE induction can be performed with other drugs or protocols; this variability should be acknowledged.
Response 4: Thank you. This has now been clarified, with a citation of the European position paper on DISE.
Comment 5: Line 428: Mixing moderate and severe patients in some analyses limits interpretation and should be explicitly stated as a limitation.
Response 5: Thank you for this valuable comment. This has been clarified using the following statement: “Combining these subgroups might affect the results and thus should be taken into account as a limitation during interpretation of statistical analyses. Furthermore, the small sample size limits the ability to draw definitive conclusions regarding collapse at the epiglottis, as only 13 of 69 participants showed epiglottic collapse at baseline. This restricted statistical power to analyze primary versus secondary epiglottic collapse.” (lines 444-449). Furthermore, the limitation state that “The relatively small sample size of 69 participants, including only seven participants with severe OSA (AHI 30/h to 50/h), may limit the generalizability of the findings, especially when comparing the results between severity subgroups.” (lines 433-436)
Results and interpretation:
Comment 6: The finding of no effect at the epiglottis is important but incomplete: the authors should distinguish between primary collapse (not resolved by MAD) and secondary collapse (may improve).
Response 6: Thank you for this very important suggestion. Unfortunately, due to the small sample size of patients with epiglottis collapse, our current study does not have adequate power to assess this difference. We have added this limitation to the discussion with the following statements: “Although no distinction was made between primary and secondary epiglottic collapse in the current study due to the small sample size, the findings at the epiglottis are in line with the findings by Fernández-Sanjuán et al. (2024).” (lines 369-372), and “Furthermore, the small sample size limits the ability to draw definitive conclusions regarding collapse at the epiglottis, as only 13 of 69 participants showed epiglottic collapse at baseline.“ (lines 446-448).
Comment 7: No significant correlation was found between changes in collapse and polysomnographic parameters (AHI, ODI, SaOâ‚‚). This point deserves a deeper discussion.
Response 7: Thank you for this insightful comment. We agree that the lack of significant correlation between changes in collapse and polysomnographic parameters warrants further reflection. This point is addressed in the Discussion (lines 385-401), and we have now expanded the paragraph to emphasize that while DISE provides valuable anatomical and functional insights, its predictive value for treatment outcomes remains limited when used in isolation (394-396).
Conclusion:
Comment 8: The authors currently conclude that MAD reduces collapse at all levels except the epiglottis and may be recommended as a first-line treatment in moderate and severe OSA patients. However, this conclusion does not fully reflect their results, which show that residual collapses persist despite MAD use, that epiglottic collapse may even worsen in some cases, and that the elimination of collapses during DISE does not necessarily translate into improvements in polysomnographic parameters. A more accurate conclusion would be:
MAD therapy effectively reduces the degree of airway collapse across most anatomical levels in participants with OSA, except at the level of the epiglottis. While these results confirm the therapeutic value of MAD, they also emphasize the need to identify patients with residual or worsening collapse who may require combined or alternative treatments. Therefore, MAD should be considered an effective therapeutic option, particularly in carefully selected patients across the OSA severity spectrum, rather than a universal first-line treatment.”
Response 8: Thank you for this thoughtful suggestion. We agree that the conclusion should accurately reflect both the strengths and limitations of our findings. The primary aim of our study was to assess the effect of MAD therapy on the degree of upper airway collapse in the total study population. Analyses across different OSA severity categories were exploratory and not powered to support definitive subgroup recommendations. In response to your comment, we have revised the conclusion to better align with the current data. We also clarified the main aim of the study in the abstract to avoid overinterpretation of subgroup findings.
Reviewer 2 Report
Comments and Suggestions for Authors
Thank you for considering me for reviewing this manuscript entitled: "Effect of mandibular advancement device treatment on the site-2 specific degree of upper airway collapse during drug-induced 3 sleep endoscopy"
This is a good quality manuscript
These are my suggestions:
Introduction: I'd reduce the introductive paragraph on OSAS definition and related comorbidities (lines 1 - 54 --> too many for obvious concepts, I'd synthesise);
Line 75 --> I would rather say that your aim is assessing the effectiveness of MAD in different anatomic patterns of collapse during DISE.
Methods: Why have not you used traditionally accepted classifications of collapse? (es. VOTE/NOHL systems) this would have increased the reproducibility of the study. Sensitivity analyses with such systems would be adequate
Please provide confidence intervals rather than 1-2 quartiles for Odds Ratios
Line 177 --> "for" repeated
Results: Do not include part of your results in the table legends (es. lines 215-216)
Discussion: Coherent with your results, I have nothing specific to suggest (line 409 --> correct the citation)
Author Response
Introduction:
Comment 1: I'd reduce the introductive paragraph on OSAS definition and related comorbidities (lines 1 - 54 --> too many for obvious concepts, I'd synthesise);
Response 1: Thank you for this comment. We have synthesized the introduction to enhance focus of our manuscript.
Comment 2: Line 75 --> I would rather say that your aim is assessing the effectiveness of MAD in different anatomic patterns of collapse during DISE.
Response 2: Thank you for this suggestion. We have edited our aim in the introduction in order to clarify our primary aim of assessing the effect of MAD on reducing degree of collapse.
Methods:
Comment 3: Why have not you used traditionally accepted classifications of collapse? (es. VOTE/NOHL systems) this would have increased the reproducibility of the study. Sensitivity analyses with such systems would be adequate.
Response 3: Thank you for this comment. The scoring system used in this paper is a modified version of the VOTE classification, taking the hypopharynx into consideration. This has now been clarified in the Methods (section 2.1, lines 124-125). Furthermore, additional sensitivity analyses using the original VOTE classification have been performed and added the online supplement.
Comment 4: Please provide confidence intervals rather than 1-2 quartiles for Odds Ratios
Response 4: Thank you for pointing this out. We have revised all tables, figures and supplementary materials to include 95% CI. Thank you for this suggestion.
Comment 5: Line 177 --> "for" repeated
Response 5: Thank you for pointing this out. This has been corrected.
Results:
Comment 6: Do not include part of your results in the table legends (es. lines 215-216)
Response 6: Thank you for this suggestion. The description on patients with missing data has now been removed from the table legend and is now only described in the results paragraph.
Discussion:
Comment 7: Coherent with your results, I have nothing specific to suggest (line 409 --> correct the citation)
Response 7: Thank you for this review and for pointing out the faulty citation, this has been corrected.
Round 2
Reviewer 1 Report
Comments and Suggestions for Authors
All suggestions have been amended.
However, a small correction should be done:
In the revised version, the authors state in lines 375-377:
"While a simulation bite mimics the position of an MAD it has a significantly lower retention and requires external force to maintain its position during DISE, which may influence the observed collapse patterns."
When using the simulation bite (SAM), we do not require any external force. Rather, it is a titrable device without tension, as opposed to traditional George gauge.
Please, re-write that sentence.
Author Response
Comment 1: In the revised version, the authors state in lines 375-377:
"While a simulation bite mimics the position of an MAD it has a significantly lower retention and requires external force to maintain its position during DISE, which may influence the observed collapse patterns."
When using the simulation bite (SAM), we do not require any external force. Rather, it is a titrable device without tension, as opposed to traditional George gauge.
Please, re-write that sentence.
Response 1: Thank you for your comment and pointing this out. The sentence has now been re-written to: "While a simulation bite is titratable and mimics the position of an MAD, it has a lower retention during DISE, which may influence the observed collapse patterns."